# Quality Assessment of Three Types of Drinking Water Sources in Guinea-Bissau

**DOI:** 10.3390/ijerph17197254

**Published:** 2020-10-04

**Authors:** Aducabe Bancessi, Luís Catarino, Maria José Silva, Armindo Ferreira, Elizabeth Duarte, Teresa Nazareth

**Affiliations:** 1Nova School of Business and Economics, Nova University of Lisbon, Campus de Carcavelos, Rua da Holanda, n.1, 2775-405 Lisbon, Portugal; 2Centre for Ecology, Evolution and Environmental Changes (cE3c), Faculty of Sciences, University of Lisbon, Campo Grande, 1749-016 Lisbon, Portugal; lmcatarino@fc.ul.pt; 3Plant-Environment Interactions & Biodiversity Lab (PlantStress&Biodiversity), Linking Landscape, Environment, Agriculture and Food Unit (LEAF), Institute of Agronomy ISA, University of Lisbon, Tapada da Ajuda, 1349-017 Lisbon, Portugal; mariajosepsantos@gmail.com; 4National Laboratory of Public Health, National Institute of Public Health (INASA), Avenida Combatentes da Liberdade da Pátria, Bissau 1004, Guinea-Bissau; armindoferreira2611@gmail.com; 5Department of Sciences and Engineering of Biosystems, Institute of Agronomy ISA, University of Lisbon, Tapada da Ajuda, 1349-017 Lisbon, Portugal; eduarte@isa.ulisboa.pt; 6Global Health and Tropical Medicine, Institute of Hygiene and Tropical Medicine, Nova University of Lisbon, Rua da Junqueira 100, 1349-008 Lisbon, Portugal; teresa.lobo.nazareth@gmail.com

**Keywords:** West Africa, water quality, *E. coli*, physicochemical, microbiological

## Abstract

The lack of access to safe drinking water causes important health problems, mainly in developing countries. In the West African country Guinea-Bissau, waterborne diseases are recognised by WHO as major infectious diseases. This study analysed the microbiological and physicochemical parameters of drinking water in the capital Bissau and its surroundings. Twenty-two sites belonging to different water sources (piped water, tubewells and shallow wells) were surveyed twice a day for three weeks, in both dry and wet seasons. Most of the microbiological parameters were out of the acceptable ranges in all types of water and both seasons and tended to worsen in the wet season. Moreover, in Bissau, the levels of faecal contamination in piped water increased from the holes to the consumer (tap/fountain). Several physicochemical variables showed values out of the internationally accepted ranges. Both well sources showed low-pH water (4.87–5.59), with high nitrite and iron levels in the wet season and high hexavalent chromium concentration in the dry season. The residual chlorine never reached the minimum recommended level in any of the water sources or seasons, suggesting a high risk of contamination. Results reveal a lack of quality in the three water sources analysed, coherent with the high number of diarrheal cases in the country. There is an urgent need to improve sanitarian conditions to reduce the disease burden caused by these waterborne illnesses.

## 1. Introduction

The access to safe drinking water should be of main concern in any society since water is a basic need for human development, health and well-being [1], but approximately 1.1 billion people in rural and peri-urban communities of developing countries do not have that access [2]. In rural areas, the lack of adequate safe water and sanitary infrastructures leaves millions with water of doubtful quality, increasing the harshness of daily life [3,4]. The mortality from diarrhoea-related diseases worldwide extents to 2.2 million people each year [5,6,7]. With a rapidly growing global population, increasing environmental degradation and the multifaceted impacts of climate change, water demand is expected to increase dramatically by nearly one-third in all major use sectors by 2050 [8,9,10]. In sub-Saharan Africa, the situation (water quantity and quality) is particularly acute due to global warming, the expansion of the Sahara Desert, civil unrest and poor governance, population growth, migration and poverty [11]. In Guinea-Bissau, West Africa, one of the poorest countries in the world according to the Human Development Index, ranking 178th among 189 countries in 2018, life expectancy at birth is 58 (male) and 61 years (female), with over 50% of the population living below the poverty line, and diarrhoea diseases being the third leading cause of premature death in 2016 [12,13,14]. Health infrastructures are poor and were greatly affected by the civil war in the late 1990s. For the majority of the rural population, the only source of water for daily needs, including drinking water, is shallow hand-dug wells (51%). No public wastewater treatment is available in the country and in 2016, the access to improved water sources was limited to 53% of the population, and only 19% had access to piped water [15]. Just Bissau and Bafatá cities currently have piped water distribution systems that, however, do not cover the entire population of these cities, particularly in the most peripheral areas. According to [3], each person has only 21 L of water for daily personal needs in Guinea-Bissau. This value is well below the 50 L minimum water requirement for human domestic use [16]. In 2019, 702,974 cases of diarrhoea diseases were reported in Guinea-Bissau [17]. Outbreaks of cholera from contaminated water through food and drinking water are common. Between 1994 and 2013, 83,635 cases and 1895 deaths from cholera were reported in the country. These epidemics occurred mainly during the wet season and lasted for more than six months. Cholera cases were reported in Bissau city (1996, 2002, 2005, 2012), Bijagós islands (1994, 2004) and Tombali (2007, 2013), which are all coastal regions, and this disease is considered endemic in the country [18,19]. However, there have not been documented outbreaks since 2013. The current study aimed to assess the quality of drinking water in Bissau city and its surroundings, and Quinhámel (hotspots of waterborne diseases such as diarrhoea including cholera) during the dry and wet seasons, and also to assess the potential correlation between distribution systems and waterborne diseases.

## 2. Materials and Methods

### 2.1. Study Area

In 2018, the population of Guinea-Bissau was 1.87 million, with an annual growth of about 2.5% [20]. The climate is tropical sub-humid, with a mean annual temperature of 26.5 °C and two seasons (dry and wet): a dry season (December to May) with north-easterly Harmattan winds and a monsoonal-type rainy season (June to November) with south-westerly winds. The most representative soil groups in the country are Ferrallisols, Plinthosols, Gleysols, Fluvisols and Arenosols [21]. A more detailed characterisation of Guinea-Bissau is presented elsewhere [22]. The cities of Bissau (including both urban and peri-urban areas) and Quinhámel (rural area) were selected for the present study (Figure 1) because they are hotspots of waterborne diseases.

Bissau has a total area of 77.5 km^2^ and a population of c. 350,000 inhabitants, where just 13% of the population has access to the piped water distribution system. The piped water system is based on a set of holes in different locations in the city, each with a pump and a tank with an upstream chlorine dosing system. Each reservoir supplies one of the five distribution areas in the city (DA): Hospital 3 de Agosto (DA1); Bandim (DA2); Hospital Central Simão Mendes (DA3), Queije (DA4) and Hospital Santo Egidio (DA5). In the peri-urban areas of Bissau and rural areas such as Quinhámel with a surface of 451 km^2^, 37 km away from Bissau [23], water for human consumption is obtained in two main ways: deep tubewells with hand-pumps and open shallow wells in which water is extracted with a bucket. 

### 2.2. Location of the Sampling Points and Sample Collection

Six water samples were collected from each of the 22 different points (Figure 1) representing three types of drinking water sources available in Bissau and Quinhámel: piped water distribution systems (piped water, urban area), deep tubewells with hand-pumps (tubewells, rural area) and shallow hand-dug wells (shallow wells, peri-urban area). In the urban area, the distribution system was sampled at four key segments: (i) hole, (ii) reservoir outlet, (iii) tap (people’s houses) and (iv) fountains (Table 1). In the peri-urban area, which lacks piped water, samples were collected from the 4 major shallow wells, which supply most of the population. In the rural area, samples were collected from the 5 major tubewells, which supply less than half of the population with water for domestic use. A total of 132 water samples were obtained in the dry (Abril and May) and wet (August and September) seasons of 2019. The collection was performed twice a day (at 7 am and 1 pm) at each sampling point, once a week for three weeks, following the collection guidelines [24]. The exact position of each sampling point was obtained through GPS (Garmin GPSMAP 64s) and mapped using GIS software (QGis 3.10 Girona). Water samples were collected using 500 mL plastic sterile flasks. After collection, they were refrigerated with locally produced shredded ice and transported to the laboratory for analysis within 6 h.

### 2.3. Analytical Procedures

All the equipment used belongs to the National Laboratory of Public Health (LNSP) in Bissau. The consumables needed for bacteriological analyses were acquired in Portugal.

#### 2.3.1. Physical and Chemical Parameters

Temperature (T), electrical conductivity (EC), turbidity, dissolved oxygen (DO), oxidation–reduction potential (ORP), salinity, total dissolved solids (TDS) and pH were measured in situ (sampling point), using a multiparameter meter (HI9829, Hanna Instruments, Woonsocket, RI, USA). Nitrate (NO_3_^−^), nitrite (NO_2_^−^), hexavalent chromium (CrVI), iron (Fe^2+^), sulphates (SO_4_^2−^), total phosphorus (TP), total alkalinity (TA), copper (Cu^2+^), sulphite (SO_3_^2−^), hardness and residual chlorine (RC) were assayed with Palintest water analysis kits (Photometer 7100, Palintest Instruments, Halma Company, UK) according to the standard methods supplied by the manufacturer (https://www.hanna.pt and https://www.palintest.com/).

#### 2.3.2. Bacteriological Analysis

Subsamples for enumeration of faecal coliforms (FC), intestinal enterococci (IE) and *Vibrio* spp. were filtered onto sterile cellulose nitrate membranes (0.22 μm pore size, 47 mm diameter, GE Healthcare Life Sciences, Little Chalfont, UK), through a hand-pump, placed in Chromogenic Coliform Agar (Biokar diagnostics, Fr), Slanetz-Bartley Agar (Oxoid, Waltham, MA, USA) and (*Vibrio* Chrome agar medium, Fr) plates, respectively, and incubated at 44.5 °C for 24 h (FC) or 48 h (IE), or 37 °C for 24 h (*Vibrio* spp.). Aerobic mesophilic microorganisms (AMM) were determined according to ISO 4833: 2003, using the incorporation technique, by pipetting 1 mL of sample onto each plate with the addition of mFc-agar Yeast Extract Agar (Biokar diagnostics, Fr), and incubation at 37 °C for 24 h. For FC, *E. coli* and IE, a volume of 100 mL was used, while for *Vibrio* spp., it was 300 mL. Typical colonies were counted, and the result was expressed as colony forming units (CFU/100 mL).

### 2.4. Statistical Analysis

Spatial and seasonal statistically significant differences among samples were evaluated by one-way ANOVA analyses of variance, followed by post hoc Tukey honestly significant difference (HSD) multi-comparison tests, using Statistica 8.0 (Stat Soft, Tulsa, OK, USA). 

Euclidean distances, principal components and classification analysis (PCA) were used to assess relationships between the different microorganism species and physicochemical parameters. The physicochemical parameters were chosen based on the eigenvalues test and the co-correlation among multiple variables, using the software Statistica 8.0 (Stat Soft, Tulsa, OK, USA). The significance level used for all tests was 0.05. In the absence of water quality standards for Guinea-Bissau, the Nigerian standard for drinking water quality (NSDWQ) [25], the European Council Directive 98/83/EC [26] and the WHO recommendations [27] were adopted in this study. Table 2 and Table 3 shows the skeleton ANOVA that includes the expected mean squares and variable under analysis.

## 3. Results

The water quality was analysed seasonally for each of the 22 sampling points representing three water source types: piped water, tubewells and shallow wells. The main microbiological and physicochemical results are described below. 

### 3.1. Physicochemical Parameters

Most of the physicochemical parameters varied according to the type of water source and the season (*p* < 0.05). The majority of the values were above the acceptable limits for drinking water, with the higher values found in the wet season (Table 4), particularly in tubewell and shallow well samples. For parameters such as DO, ORP, TP and alkalinity, no established ranges are available to allow a comparison. The pH of piped water was within the acceptable range for human consumption in both seasons, contrasting with the water samples from shallow wells and tubewells, always acidic (pH 4.9–5.6) and clearly below the recommended limit of 6.5. In shallow wells, the turbidity values varied from 14.43 in the dry season to 19.96 NTU in the wet season, well above the maximum value of 5 recommended by NSDWQ. In most shallow wells, the water is taken out with buckets, which increases the turbidity. Nitrite exceeded the maximum recommended levels of 0.50 (WHO) and 0.2 mg L^−1^ (EU and NSDWQ) in the wet season, for the three water sources analysed. Piped water in Bissau averaged 0.62 mg L^−1^, while extremely high nitrite concentrations of 2.84 and 4.66 mg L^−1^ were found in tubewell and shallow well waters, respectively. The levels of hexavalent Cr(VI) exceeded the limits proposed by WHO and NSDWQ, in the dry season, for the three water sources, with shallow wells revealing extremely high values up to 0.37 mg L^−1^ in the wet season. The recommended maximum iron content is 0.2 mg L^−1^ L (WHO and EU), but higher values were observed in the wet season in tubewell (1.80 mg L^−1^) and shallow well (3.51 mg L^−1^) water samples. The residual chlorine never reached the levels recommended for drinking water by WHO and the EU in either season or water source analysed. Further, the values found for piped water in Bissau were not different from the ones for untreated water (see Appendix A. Raw data tables).

### 3.2. Microbiological Assessment

#### 3.2.1. Microbiological Quality of the Three Sources of Drinking Water: Piped Water, Tubewells and Shallow Wells

For the three drinking water sources available to the population, the results showed that the majority of the microbiological variables failed to meet the drinking water standards recommended by WHO and the EU; the only exception was IE in piped water, during the dry season, and in tubewells, during the wet season (Table 5). Overall, wells showed higher values than piped water in Bissau, in both seasons, although shallow wells exhibited very high microbial contamination in both seasons. No significant differences (*p* > 0.05) were detected between the two main types of water delivery in Bissau, public fountain and house tap. Despite the higher AMM concentration observed in piped water, the FC in both seasons was lower than in tubewells. FC, *E. coli* and AMM were generally higher during the wet season in all the water sources available for human consumption. Only IE detected in tubewells and shallow wells increased in the dry season, with high values observed in shallow wells (see Appendix A. Raw data tables).

#### 3.2.2. Piped Water Distribution Systems

Table 5 presents the values of microbiological parameters recorded in the dry and wet seasons, according to the segments of the piped water distribution system: pre-consumer water (hole, reservoir outlet) and consumer water (house tap, public fountain). 

In all the sampled points, quality varied with the season (*p* < 0.05) with the wet season presenting the worst condition. FC, IE and *E. coli* were present in all the sampling points, with exception of the holes. However, AMM detected in both seasons exceeded the WHO and EU acceptable limits for drinking water and values tended to increase significantly in the wet season. Overall, the reservoir outlets, taps and fountains were the sampling points with the highest contamination levels, and the values for consumer water are not significantly different. It is important to notice that all the microbiological parameters appeared better in the dry season (*p* < 0.05) than in the wet season and that the microbiological quality of water decreased from the source to the final consumer. In addition, all the recorded values were beyond the acceptable limits for drinking water according to the EU and WHO. IE, *Vibrio cholerae, V. parahaemolyticus* and *V. vulnificus* were not detected in any sample (see Table 3).

#### 3.2.3. Comparison between Piped Water Distribution Areas in Bissau

In Table 6, it is possible to compare the microbiological parameters between the five distribution areas (DA) of Bissau city. The majority of the values exceeded the maximum recommended by WHO, in both seasons, despite the trend of worse microbiological water quality in the wet season. The measured values only complied with the international standards in the case of *E. coli* during the dry season at DA1. 

DA1 is the distribution area with lower values of microbiological contamination in the dry season but they greatly increase in the wet season, namely for AMM. DA1 was the least contaminated system, whereas the other four sampled distribution areas presented similarly high FC levels. Three of the four bacteriological indicators showed higher counts than the acceptable limits for drinking water in all samples. For the three analysed microbiological parameters in both seasons, there were significant differences (*p* < 0.05) between the distribution areas (Table 6).

#### 3.2.4. Correlation/Relationship between Physicochemical Parameters and Microorganism Abundance

The possible interaction between physicochemical variables was analysed through principal components analysis (PCA) based on Euclidean distance. Figure 2 shows the PCA plots showing the relationship between those parameters for each of the three water sources studied in the dry and wet seasons. For piped water, in the dry season, it seems that the development of faecal coliforms and AMM can be associated with high NO_2_^−^ levels as well as by high Cr(VI) levels, while for the wet season, the multivariate analysis suggests that the FC concentration can be associated with the levels of SO_3_^2−^ as well as NO_2_^−^ and NO_3_^−^. TDS and AMM, whose concentrations increase with pH, probably due to a greater number of bacteria, can survive at a higher pH. Concerning tubewells’ water, in the dry season, results suggest a link between AMM development and the levels of TP, SO_4_^2−^ and Fe^2+^, and no relation between nitrogen compounds (NO_2_^−^ and NO_3_^−^), while in the wet season, FC seems to be related to the levels of particles in suspension (TDS and turbidity). AMM is correlated with Fe^2+^, Cr(VI) and NO_3_^−^, while *E. coli* is correlated with NO_3_^−^. For shallow wells in the dry season, the results suggest a relationship between all the microbiological parameters and EC, TDS and SO_3_^2−^ but also NO_2_^−^ and SO_4_^2−^, while in the wet season, all but AMM microbial composition parameters, in particular, show a direct relation with Fe^2+^, SO_4_^2−^ and TP, but also with SO_3_^2−^, NO_2_^−^ and TA.

In an overall appreciation, the stronger and more frequent relationships between physicochemical and microbiological parameters were found between faecal coliforms (including *E. Coli)* and IE with NO_2_^−^, NO_3_^−^, SO_3_^2−^ and SO_4_^2−^ and the variables linked to the particles in suspension (TDS and turbidity).

## 4. Discussion

Since there are no guidelines for drinking water in Guinea-Bissau, the key water quality parameters were compared to the recommendations from the European Union [26], World Health Organization [27] and Nigerian [25] guidelines for drinking water. The herein recorded values for physicochemical and microbiological parameters confirm the critical situation of this resource in Guinea-Bissau. pH values from the piped water were within the acceptable limits in both seasons. Similar results have already been described by other authors [29,30]. Regarding the physicochemical parameters, all the samples from tubewells and shallow wells were acidic (pH values between 4.89 and 5.59), outside the suitable range for drinking water. Only one sampling point (rural L14) presented an acceptable pH value, during the wet season, according to the above standards. Increased pH (of L14) can only be satisfactorily explained by groundwater discharge and infiltration due to precipitation.

Values as low as pH 3.98 were found at the peri-urban sampling point L8 in the wet season, probably due to the acidity of soils, a major constraint in West Africa [29,31]. Such low pH values have been considered responsible for dental erosion [32,33,34] which represents a real problem in a country where the healthcare system is fragile and not universal. In addition, it causes problems at the stomach level, and since the pH of the water is acidic, the bacteria that are there will be adapted to this condition and potentially the acidic barrier of our stomach will be more easily overcome by pathogens and cause disease. Water temperature averaged 28.6 °C, above the recommended 25 °C maximum for drinking water according to the EU [26], WHO [27] and NSDWQ [25]. Such a high water temperature can promote microbial growth and chemical reactions [35,36]. In all water sources in both seasons, Cr(VI) contents were beyond the acceptable limits (WHO and EU standards); this element is a human carcinogen via inhalation [37]. Cr(VI) has high environmental mobility and can originate from anthropogenic and natural sources and also can be linked to TDS. 

According to the Nigerian standard, all water sources had values above the recommended, except piped water and tubewells in the wet season. Iron (Fe^2+^) contents were above the recommended values for drinking water, which may be related to the rain-promoted leaching of acidic and iron-rich soils (in general Ferralsols) common across the country. Other parameters, mainly NO_3_^−^, hardness, EC, SO_4_^2−^, total phosphorus, SO_3_^2−^, Cu^2+^, total alkalinity and residual chlorine concentrations, were below the parametric values for drinking water in both seasons [25,26,27] in most of the sampling points. However, the three water types showed increased concentrations of NO_2_^−^ in the wet season, a possible result from wastewater and garbage dumps that might infiltrate wells and contaminate the water. Not surprisingly, since it is very soluble and does not bind to soils, NO_2_^−^ has a high potential to migrate to groundwater. Therefore, shallow wells and tubewells presented much higher levels than piped water, which is not in contact with the upper parts of the soil. 

Such a high concentration of NO_2_^−^ has also been described as responsible for the risk of methemoglobinemia in children, and toxic to humans as well as to animals [38,39,40]. The same result has been found in other studies [29,30,41]. Overall, the quantifications of NO_2_^−^, NO_3_^−^, SO_3_^2−^, SO_4_^2−^ and turbidity can give a good idea of the microbial contamination of the water. All of these variables’ results are from human activity or the season, in the case of turbidity. Residual chlorine contents were below the acceptable limits (WHO and EU standards). As chlorine is considered a water disinfectant, the shortage of chlorine in drinking water may influence microorganism population development. Turbidity is low in the dry season and tends to increase in the rainy season. In addition, in the dry season, the volume of water from the wells is very low, which leads to resuspension when the water is removed with the bucket. In the wet season, turbidity is a consequence of the infiltration and percolation of the surface. As shown by [42], turbidity is related to gastrointestinal diseases’ drinking water-related risks. So, evaluating the impact of drinking water turbidity on the incidence of gastrointestinal diseases could represent a useful tool to support water suppliers’ decisions.

Microbiological results showed that all the sampled water sources were grossly polluted with FC, with values higher than those recommended by the European Union, World Health Organization and Nigerian guidelines. Faecal contamination was detected in the piped water system after the holes, increasing from the reservoir to the house taps, and from these to fountains. These results can be explained by the fact that in Bissau, the supply system consists of a water source (hole) from which water is pumped into a reservoir mounted a few meters from the hole. Upstream of the reservoir, there is a chlorine dosing system, where the amount of chlorine is calculated according to the water flow rate and is injected into the reservoir to promote water disinfection and subsequently distributed to the population. There is no other pre-treatment system or water filtration before or after the water enters the reservoirs and distribution system. However, at the time of sampling, the entire dosing system was inactive.

In addition, the current piped water distribution systems were built during the colonial period and the degradation of water distribution systems favours microorganism growth [3]. As *E. coli* and FC were not detected in any of the holes in either season (see Table 3), the differences between DA can be attributed to the quality of water treatment and/or to different degrees of degradation of the piped water distribution network. Although shallow wells in peri-urban areas of Bissau are protected from free access by animals, most of them lack wellheads. Therefore, contamination by FC is not completely prevented and occurs throughout the year, worsening in the wet season, as shown by the recorded higher levels of contamination. This increase can be attributed to the facilitated mobility of this type of pollutant by rainwater infiltration and percolation [3,43,44], due to the type of soil and water well structure. The proximity of latrines (≤30 m), whose influence has been previously described [45,46,47], and the presence of freely wandering domestic animals and the contact of the well bucket, plastic buckets (used to withdraw water from shallow wells) and rope with contaminated soil contribute to the situation [48]. Indeed, the rainwater infiltration and percolation, facilitated by the sandy nature of the soil, were the probable driving forces for the mobilisation and consequent increment of presumptive faecal coliforms counts found in the wet season [3,43,49]. IE was not detected in piped water in both seasons, but wells were contaminated with IE. AMM are those that were detected in large quantities at all sampling points with a significant increase in the wet season. Although many studies report the presence of *Vibrio cholerae* in Guinea-Bissau [50,51,52], this species was not detected in this study. In addition, no cases of cholera have been reported in Guinea-Bissau since 2013, which may corroborate the results. On the other hand, since only culture media were used and these bacteria can enter a viable but non-cultivable state (VNC) when environmental conditions are not the most favourable, this can be a limitation. Another limitation of the current study was the impossibility of access to all the segments of the piped water system in all the distribution areas, which hinders the correct comparison between the five distribution systems. Furthermore, the study was restricted to Bissau and Quinhámel, in order to ensure a 6-h distance to the field laboratory, which reduced the geographic representativeness of our sampling. 

## 5. Conclusions

All the studied/analysed sources of water available for human consumption in Guinea-Bissau were contaminated. The highest contamination was observed in shallow wells, followed by tubewells and piped water. In piped water, the highest contamination was found in fountains, followed by taps and reservoir outlets. Contamination of water sources varied with the season, increasing in the wet season for most of the parameters. Two main types of water distribution systems can be found: a centralised distribution chain in Bissau and single-point systems present in the peri-urban and rural areas. Water quality in these two systems must be addressed differently. In Bissau, water quality seems good enough at the sources (holes) but an ineffective treatment and an old and damaged distribution system contribute to delivering water of poor quality. Data highlighted the need for urgent care of all of the water chain supply, namely the reservoir outlets, taps and fountains, and to raise the general awareness about water contamination. The single-point systems (tubewells and shallow wells) represent a different problem. Water is consumed directly, without previous treatment and, in most cases, it is contaminated by sewage, particularly in the wet season. 

Further, some chemical properties, namely pH and iron content, show unsuitable values for human consumption. Based on these results and given the high incidence of diarrhoea among young children in Bissau and Quinhámel, we suggest the implementation of several feasible procedures by the competent authorities to optimise water use: for piped water, these are chlorine dosing to promote water disinfection, periodical flushing and disinfection of the reservoir, starting planning for the replacement of the whole water distribution system in the basement area and promoting periodic analyses of the water consumed by the population in order to assess its quality; for deep tubewells with hand-pumps and shallow hand-dug wells, these include removing nearby latrines and garbage dumps and promoting studies on domestic disinfection practices, such as adding inexpensive *Moringa oleifera* extract or powder to drinking and kitchen water, according to internationally accepted practices. In addition, the water sources can be managed and protected based on national legislation and associated international conventions (e.g., the water convention: responding to global water challenges). Sustainable management of water is still a great concern across the country, particularly in rural areas. Highlights for water management are the preservation of water, its sustainable use and integration of the local stakeholders and community in the overall design of management initiatives. 

Further research should focus on specific measures to improve the quality of the water consumed in Guinea-Bissau, and also to evaluate the conditions of water infrastructures. Furthermore, as there is no treatment for wastewater and hospital waste, it may be important to assess the risk of water contamination by numerous pollutants that are discharged in these waste streams that can be acutely problematic for human health, such as drugs or heavy metals.

## Figures and Tables

**Figure 1 ijerph-17-07254-f001:**
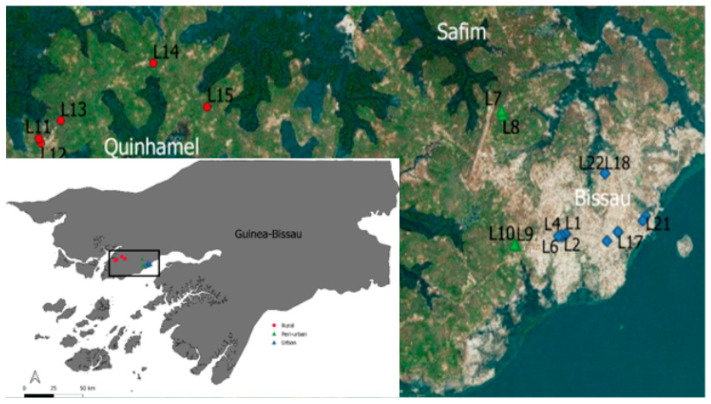
Location of the sampling points in Bissau and Quinhámel: urban (L1, L2, L3, L4, L5, L6, L16, L17, L18, L19, L20, L21, L22), peri-urban (L7, L8, L9, L10) and rural (L11, L12, L13, L14, L15).

**Figure 2 ijerph-17-07254-f002:**
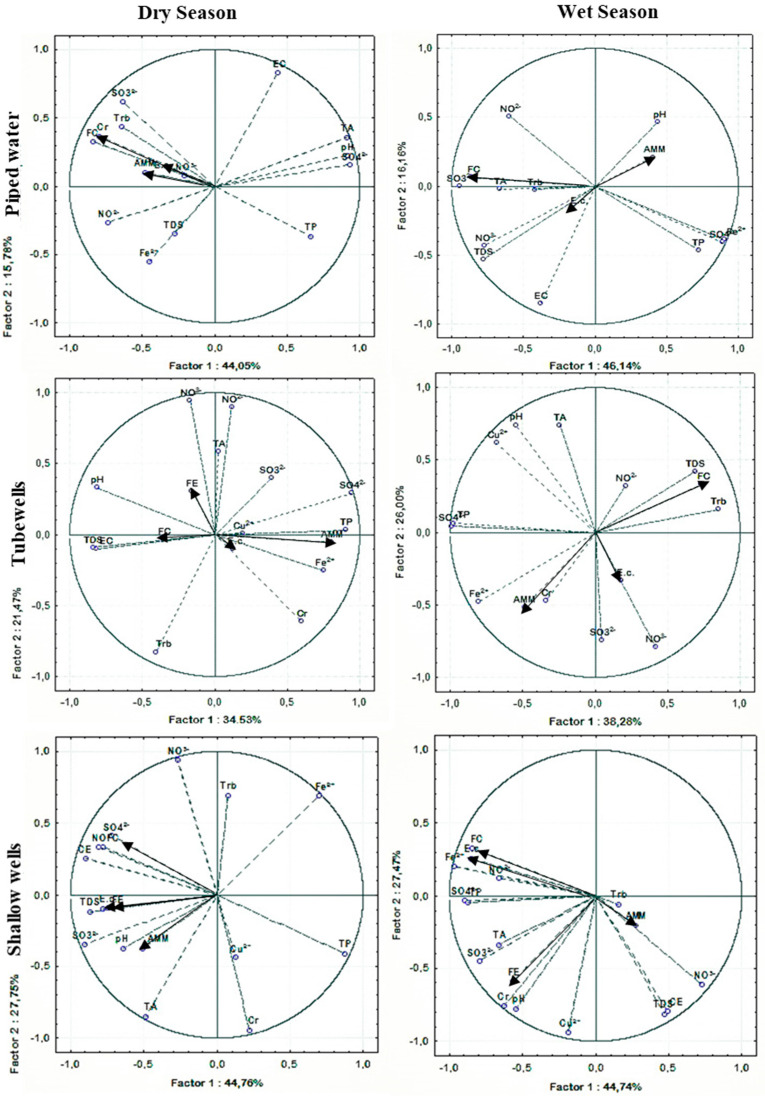
PCA plots based on Euclidean distances showing the relationship between microbiological and physicochemical parameters for piped water, tubewells and shallow wells in the dry and wet seasons. The microbiological parameters are indicated by vectors; only active variables were considered. Trb—turbidity; TA—total alkalinity; E. c.—*Escherichia coli*; AMM—aerobic mesophilic microorganisms; IE—intestinal enterococci; FC—faecal coliforms TP—total phosphorus; TDS—total dissolved solids.

**Table 1 ijerph-17-07254-t001:** Sampled segments of the piped water of each of the five distribution systems in Bissau (urban area).

Name	Distribution Area	Sampled Segment
Hole	Reservoir Outlet	Tap	Fountain
DA1	Hospital 3 de Agosto	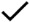	-	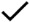	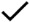
DA2	Bandim	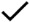	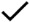	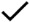	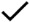
DA3	Hospital Central Simão Mendes	-	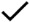	-	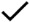
DA4	Queije	-	-	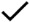	-
DA5	Hospital S. Egidio	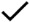	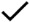	-	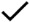

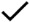
—sampled; - not sampled.

**Table 2 ijerph-17-07254-t002:** Skeleton analysis of variance for Table 5—season × distribution areas.

ANOVA Univariate Tests of Significance Effective Hypothesis Decomposition					
	DF	FC	*E. coli*	FE	AMM
**Intercept**	1	0.000 ***	0.000 ***	0.000 ***	0.000 ***
**Season**	1	0.000 ***	0.091 ***	0.091 ns	0.000 ***
**Distribution Area**	4	0.000 ***	0.000 ***	0.000 ***	0.000 ***
**Season × Distribution Area**	4	0.901 ns	0.401 ns	0.401 ns	0.002 **
**Error**	146				

DF—degrees of freedom; FC—faecal coliforms; FE—faecal enterococci; AMM—aerobic mesophilic microorganisms; ** *p* ≤ 0.01; *** *p* ≤ 0.001; ns—not significant.

**Table 3 ijerph-17-07254-t003:** Skeleton analysis of variance for Table 6—season × water distribution system segment.

ANOVA Univariate Tests of Significance Effective Hypothesis Decomposition					
	DF	FC	*E. coli*	FE	AMM
**Intercept**	1	0.000 ***	0.000 ***	0.000 ***	0.000 ***
**Season**	1	0.000 ***	0.024 *	0.024 *	0.000 ***
**System/segment**	3	0.000 ***	0.000 ***	0.000 ***	0.000 ***
**Season × System/segment**	3	0.091 ns	0.422 ns	0.422 ns	0.000 ***
**Error**	148				

DF—degrees of freedom; FC—faecal coliforms; FE—faecal enterococci; AMM—aerobic mesophilic microorganisms; * *p* ≤ 0.05, *** *p* ≤ 0.001, ns—not significant.

**Table 4 ijerph-17-07254-t004:** Mean values (*n* = 6) of the major water physicochemical parameters analysed during the dry and wet season at Bissau (urban and peri-urban area) and Quinhámel (rural area), Guinea-Bissau. Limits for drinking water are indicated, as recommended by WHO, the EU and NSDWQ.

Parameter	Unit	Dry			Wet			WHO/EU	NSDWQ
		Water Sources							
		Piped Water	Tubewell	Shallow Wells	Piped Water	Tubewell	Shallow Wells		
pH	-	8.02	4.89	5.59	8.23	5.46	4.87	≥6.5–≤9.5	≥6.5–≤8.5
T	°C	29.9	28.2	27.5	30.5	28.0	27.7	-	b
Salinity	ppm	0.25	0.07	0.04	0.25	0.07	0.06	-	-
Turbidity	NTU	0.84	1.91	14.43	1.56	1.95	19.96	a	<5
EC	(µs cm^−1^)	521	118	151	531	152	125	<2500	<1000
DO	mgL^−1^	3.91	4.65	5.21	0.00	0.00	1.06	-	-
ORP	mV	121.24	225.90	187.87	24.98	43.20	42.47	-	-
TDS	mgL^−1^	224.1	61.1	73.7	269.7	76.4	9	-	<500
Nitrite (NO_2_^−^)	mgL^−1^	0.01	0.01	0.02	0.62	2.84	4.66	<0.5	<0.2
Nitrate (NO_3_^−^)	mgL^−1^	0.89	0.95	4.40	0.00	0.04	0.14	<50	<50
Chromium (CrVI)	mgL^−1^	0.12	0.07	0.12	0.00	0.01	0.37	<0.05	<0.05
Iron (Fe^2+^)	mgL^−1^	0.08	0.01	0.20	0.17	1.80	3.51	<0.2	<0.3
Sulphate (SO_4_^2−^)	mgL^−1^	1.07	1.81	3.03	0.61	0.21	0.18	-	<100
Sulphite (SO_3_^2−^)	mgL^−1^	15	14	13	17	15	15	<250	-
P (TP)	mgL^−1^	13.34	0.18	0.11	0.61	0.21	0.18	-	-
Alkalinity (TA)	mgL^−1^	208.9	6.2	22.6	234.3	8.5	28.0	-	-
Copper (Cu^2+^)	mgL^−1^	0.04	0.01	0.10	0.04	0.01	0.09	<2	<1
Hardness	mgL^−1^	8.58	26.49	18.08	5.19	27.96	18.49	-	<150
RC	mgL^−1^	0.16	0.10	0.13	0.14	0.08	0.15	0.2–1	0.2–0.25

EU—European Union parametric values for drinking water [26]; WHO—World Health Organization guideline values [28]; NSDWQ—Nigerian Standard for Drinking Water Quality [25]; T—temperature; EC—electrical conductivity; DO—dissolved oxygen; ORP—oxidation–reduction potential; TDS—total dissolved solids; RC—residual chlorine; a—acceptable to consumer; b—room temperature. WHO/EU and NSDWQ—the maximum recommended values for drinking water.

**Table 5 ijerph-17-07254-t005:** Means ± standard deviations (*n* = 6) and ANOVA results for the comparison of microbiological parameters between the segments of the water distribution system in Bissau, in the dry and wet seasons.

Season	Parameter	Hole	Reservoir Outlet	Tap	Fountain	WHO/EU
Dry	*E. coli*	0 ± 0 bA	0.7 ± 0.8 aA	0.6 ± 0.8 aA	0.6 ± 0.6 aA	0
	FC	0 ± 0 bA	6.3 ± 4.2 aA	9.2 ± 6.8 aA	7.5 ± 2.6 aB	0
	IE	0	0	0	0	0
	AMM	52.9 ± 38.9 bA	138.8 ± 46.0 aB	126.2 ± 53.7 aB	156.4 ± 79.2 aB	<20
	*Vibrio* spp.	0	0	0	0	0
Wet	*E. coli*	0 ± 0 bA	1.0 ± 0.7 aA	0.8 ± 0.7 aA	1.1 ± 0.8 aA	0
	FC	0 ± 0 bA	9.0 ± 4.3 aA	12.9 ± 8.1 aA	12.7 ± 4.0 aA	0
	IE	0	0	0	0	0
	AMM	96.3 ± 63.9 bA	269.7 ± 42.0 aA	282.3 ± 35.7 aA	240.0 ± 39.7 aA	<20
	*Vibrio* spp.	0	0	0	0	0

FC—faecal coliforms; FE—faecal enterococci; AMM—aerobic mesophilic microorganisms; WHO—World Health Organization guideline values [27], EU—European Union parametric values for drinking water [26]. Mean values followed by the same letter do not differ significantly at *p* ≤ 0.05 (lower-case letter in a row and upper-case letter in a column). WHO/EU—the maximum values for drinking water recommended.

**Table 6 ijerph-17-07254-t006:** Mean values (*n* = 6) of the microbiological parameter of drinking water in each distribution area, during the dry and wet seasons in Bissau.

		Piped Water Distribution Areas					
Parameter	Season	DA1	DA2	DA3	DA4	DA5	WHO/EU
FC	Dry	3.5 cA	8.5 bcA	10.3 bA	17.5 bA	5.7 aA	0
	Wet	6.4 cA	13.6 bA	16.2 bA	22.3 aA	11.7 bA	
*E. coli*	Dry	0.0 cA	0.6 bcA	0.8 abA	1.3 aA	1.0 aA	0
	Wet	0.8 aA	1.1 aA	0.6 aA	1.3 aA	1.5 aA	
AMM	Dry	70.6 cB	105.8 bcB	259.0 aA	197.7 aA	195.5 aA	<20
	Wet	298.4 aA	289.2 aA	292.8 aA	232.2 aA	240.2 aA	
*Vibrio* spp.	Dry	0	0	0	0	0	0 0
	Wet	0	0	0	0	0	

DA1—Hospital 3 de Agosto; DA2—Bandim; DA3—Hospital Central Simão Mendes; DA4—Queije; DA5—Hospital Santo Egidio. FC—faecal coliforms; IE—intestinal enterococci; AMM—aerobic mesophilic microorganisms. WHO—World Health Organization guideline values [27], EU—European Union parametric values for drinking water [26]; WHO/EU—maximum values for drinking water recommended. For each pathogen and season, means followed by the same letter are not significantly different at *p* ≤ 0.05. (lower-case letter in a row and upper-case letter in a column).

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
