# Peer review of "Quality Assessment of Three Types of Drinking Water Sources in Guinea-Bissau"

_ijerph, 2020, doi:10.3390/ijerph17197254_

Round 1

Reviewer 1 Report

Overall, I found the paper to be a well designed and thought out study, and highlights the clear public health dangers posed by an unsafe drinking water supply in Guinea-Bissau. The authors did a nice job running through the study, with each section's presentation logically presented.  A few minor suggestions are below.  In addition, the authors may want to consider providing a bit more up front discussion about the lack of drinking water treatment infrastructure for the piped systems - it wasn't inherently clear where there was treatment or where there wasn't.  Hence, a few sentences on whether or not their is a drinking water treatment plant or filtration site, and the extent to which treatment is provided prior to water entering the distribution system would help better inform the readers understanding of the severity of degradation along the piped distribution system. 

Minor suggestions below.

 Line 38-39: Consider changing; “to prevent this important health problem.” “to reduce the disease burden caused by these waterborne illnesses.”  This is a more precise formulation.

Line 290: Remove “however” starting the paragraph.

Line 339: “the no cases of cholera” were found is somewhat incongruous with the introduction discussing cholera, where the frequency of outbreaks are highlighted and it is noted as being endemic.  Perhaps this can be remedied by noting this was a historic problem in the introduction, but that there haven’t been documented outbreaks since 2013.

Line 380: with no wastewater or hospital waste treatment, why limit the need for additional information to metals and drugs.  There are numerous pollutants that are discharged in these waste streams that can be acutely problematic for human health.

Author Response

Rev 1

Overall, I found the paper to be a well designed and thought out study, and highlights the clear public health dangers posed by an unsafe drinking water supply in Guinea-Bissau. The authors did a nice job running through the study, with each section's presentation logically presented. 

A few minor suggestions are below.  In addition, the authors may want to consider providing a bit more up front discussion about the lack of drinking water treatment infrastructure for the piped systems - it wasn't inherently clear where there was treatment or where there wasn't.  Hence, a few sentences on whether or not their is a drinking water treatment plant or filtration site, and the extent to which treatment is provided prior to water entering the distribution system would help better inform the readers understanding of the severity of degradation along the piped distribution system.

Answer – The reviewer is right and this information was added at the final of section 2.1.

Minor suggestions below.

 Line 38-39: Consider changing; “to prevent this important health problem.” “to reduce the disease burden caused by these waterborne illnesses.”  This is a more precise formulation.

Answer – Thank you for this suggestion and this suggestion was duly addressed in the present revision.

Line 290: Remove “however” starting the paragraph.

AnswerThank you for this suggestion, the change has been done.

Line 339: “the no cases of cholera” were found is somewhat incongruous with the introduction discussing cholera, where the frequency of outbreaks are highlighted and it is noted as being endemic.  Perhaps this can be remedied by noting this was a historic problem in the introduction, but that there haven’t been documented outbreaks since 2013.

AnswerThank you very much for the comments. Authors agreed and the suggestion was added in the Introduction and Discussion section.

Line 380: with no wastewater or hospital waste treatment, why limit the need for additional information to metals and drugs.  There are numerous pollutants that are discharged in these waste streams that can be acutely problematic for human health.

Answer – Thank you for this suggestion and this issue was addressed in the present revision.

Reviewer 2 Report

This study examined the water quality of 22 sites in Guinea-Bissau.  A spectrum of water parameters were tested.

Basically I think this study is of great important for the regional water quality control practice, and will provide valuable inference to the local government, though the scientific methods are routine. The paper is well organized and written.

Some generic comments can be found below.

The water quality was tested twice a day for three weeks. The readers may like to see the dynamic curve of the results but not just mean values.

Table 2 and 3 can be changed to figures.

The language can be slightly polished before publication.

For example, Line 28, -piped ..shallow wells--->(piped...wells),

Line 39-40, this can be deleted, as it is not conclusions but recommendation

Line 94, 22 different--->the 22

Author Response

Rev 2

This study examined the water quality of 22 sites in Guinea-Bissau.  A spectrum of water parameters were tested.

Basically I think this study is of great important for the regional water quality control practice, and will provide valuable inference to the local government, though the scientific methods are routine. The paper is well organized and written.

Some generic comments can be found below.

The water quality was tested twice a day for three weeks. The readers may like to see the dynamic curve of the results but not just mean values.

Answer – Thank you for this quite interesting suggestion, but we would like to clarify that the data were collected in two periods of three weeks each, the temporal variation was not very marked and is reflected in the standard deviations. However, taking into account the suggestion of reviewer 3, the raw data will be available in the supplementary information, which will allow more detailed analysis.

Table 2 and 3 can be changed to figures.

Answer – Thank you very much for this suggestion. We analysed several similar works which have already been published on this subject and the most common way of presenting physical-chemical and microbiological data we found was tabled form. Thus, we think it is would be better to keep the tables, which can allow easy comparison with data already published.

The language can be slightly polished before publication.

AnswerEnglish language revision was done according to the Reviewer’s advice. Changes included are highlighted in yellow colour.

For example, Line 28, -piped ..shallow wells--->(piped...wells),

Answer – The reviewer is right and the change has been made.

Line 39-40, this can be deleted, as it is not conclusions but recommendation

Answer Thank you very much for this suggestion and this change has been done.

Line 94, 22 different--->the 22

Answer – Thank you, change done.

Reviewer 3 Report

The authors report on the chemical and microbiological quality of a variety of drinking water sources in Guinea-Bissau.

The new scientific content of the paper is negligible. However, the analytical results provide a basis for improving drinking water quality in Guinea-Bissau. They provide quantitative information on the nature of the water quality problems there, which is important for decision making in the water treatment sector. This should lead to positive health outcomes.

I was interested to see the magnitude of the nitrite concentrations in wells in the wet season. These are very high and similar to the well nitrate concentrations in the dry season. Either there has been some confusion between nitrate and nitrite in the analysis and presentation of results, or full oxidation of reduced N to nitrate is not taking place in the wet season. The very low wet season ORP in the wells suggests this might be the case. It would have been interesting to see ammonium concentrations. The piped water concentration of phosphorus is exceptionally high, and, if not erroneous, is presumably due to one of the water treatment processes.  

The authors' suggestions on the causes of the identified water quality problems, and how they might be tackled, are most useful.

The upper and lower case lettering to identify significance in Tables 3 and 4 is unclear. A skeleton ANOVA table showing the factors included are needed for these analyses. The authors also need to present their raw data in tabular or graphical form.    

Author Response

Rev 3

The authors report on the chemical and microbiological quality of a variety of drinking water sources in Guinea-Bissau.

The new scientific content of the paper is negligible. However, the analytical results provide a basis for improving drinking water quality in Guinea-Bissau. They provide quantitative information on the nature of the water quality problems there, which is important for decision making in the water treatment sector. This should lead to positive health outcomes.

I was interested to see the magnitude of the nitrite concentrations in wells in the wet season. These are very high and similar to the well nitrate concentrations in the dry season. Either there has been some confusion between nitrate and nitrite in the analysis and presentation of results, or full oxidation of reduced N to nitrate is not taking place in the wet season. The very low wet season ORP in the wells suggests this might be the case. It would have been interesting to see ammonium concentrations.

Answer – The reviewer is right and we would like to thank the reviewer for this comment. In fact, there were some values exchanged, and that have already been corrected in the table. However, these were the values detected as can be seen in the supplementary information table, and the only plausible explanation for this may be due to the fact that in the wet season the dissolved oxygen values in the wells are very low as ORP as well, which suggests that we are in anoxic conditions and the oxidation-reduction process is not taking place. And this suggests that in this water there may be high amounts of ammonium (NH₄⁺), resulting from waste food, excrement, and other organic substances. And as mentioned by the reviewer, it would undoubtedly be interesting to have the ammonium values, but unfortunately taking into account the laboratory conditions at the time, it was not possible to analyze this important parameter. Please note that due to some changes, tables 2 has been changed to 4.

The piped water concentration of phosphorus is exceptionally high, and, if not erroneous, is presumably due to one of the water treatment processes. 

Answer – Thank you very much for the comment. Regarding this issue, it is important to say that there were two sampling points whose values are higher than the rest. However, the 6 repetitions of these points are always similar, and taking into account this is an average value that ends up influencing the entire mean. This peak may be the result of some contamination.

The authors' suggestions on the causes of the identified water   quality problems, and how they might be tackled, are most useful.

Answer – Tank you, this is the main objective of this work.

The upper and lower case lettering to identify significance in Tables 3 and 4 is unclear.

Answer – The reviewer is right and this change has been done. The authors would like to inform that table 3 and 4 has been changed to 5 and 6 respectively.

A skeleton ANOVA table showing the factors included are needed for these analyses.

Answer – The reviewer is right and all these changes have been included in section 2.4

The authors also need to present their raw data in tabular or graphical form.

Answer – As suggested by the reviewer, the raw data were placed as supplementary information